# Single-nuclei characterization of pervasive transcriptional signatures across organs in response to COVID-19

**The COVID Tissue Atlas Consortium, Alejandro A Granados[1], Simon Bucher[2], Hanbing Song[3], Aditi Agrawal[1], Ann T Chen[4], Tien Peng[4], Norma Neff[1], Angela Oliveira Pisco[1], Franklin Huang[3], Bruce Wang[2]***

[1]Chan-Zuckerberg Biohub, San Francisco, United States; [2]Division of Gastroenterology, Department of Medicine, University of California, San Francisco, San Francisco, United States; [3]Department of Medicine, San Francisco Veterans Affairs Medical Center, University of California San Francisco, San Francisco, United States; [4]Yale University, New Haven, United States

*For correspondence:
bruce.wang@ucsf.edu

Group author details:
The COVID Tissue Atlas Consortium See page 19

Competing interest: The authors declare that no competing interests exist.

## Abstract

**Background:** Infection by coronavirus SARS-CoV2 is a severe and often deadly disease that has implications for the respiratory system and multiple organs across the human body. While the effects in the lung have been extensively studied, less is known about the impact COVID-19 has across other organs.

**Methods:** Here, we contribute a single-nuclei RNA-sequencing atlas comprising six human organs across 20 autopsies where we analyzed the transcriptional changes due to COVID-19 in multiple cell types. The integration of data from multiple organs enabled the identification of systemic transcriptional changes.

**Results:** Computational cross-organ analysis for endothelial cells and macrophages identified systemic transcriptional changes in these cell types in COVID-19 samples. In addition, analysis of gene modules showed enrichment of specific signaling pathways across multiple organs in COVID-19 autopsies.

**Conclusions:** Altogether, the COVID Tissue Atlas enables the investigation of both cell type-specific and cross-organ transcriptional responses to COVID-19, providing insights into the molecular networks affected by the disease and highlighting novel potential targets for therapies and drug development.

**Funding:** The Chan-Zuckerberg Initiative, The Chan-Zuckerberg Biohub.

## Editor's evaluation

This important work provides a valuable data resource to study the systemic effects of severe COVID-19 infection. It further provides compelling evidence for a conserved transcriptional signature in macrophages and endothelial cells in response to COVID-19 and suggests potential molecular interactions between the two cell types. The work will be of broad interest to researchers investigating the physiological impacts of viral infection and their potential treatment.

## Introduction

COVID-19 (coronavirus disease 2019) is the most devastating infectious disease in recent history. The pandemic has impacted all parts of the globe and resulted in nearly 500 million infections and over 6,000,000 deaths (*Dong et al., 2020*). Approximately 14% of infected unvaccinated individuals develop a severe clinical disease that requires hospitalization (*Wu and McGoogan, 2020*). While the primary organ affected by severe COVID-19 is the lung, many other organs, including the heart, liver, and kidney, are also affected (*Mokhtari et al., 2020*; *Xie et al., 2022*; *Wang et al., 2021*). In addition, long-COVID has become an important and common sequela in those who recover from infection. Long-COVID often affects multiple organs and is more common in patients with a severe initial infection (*Taquet et al., 2021*).

The systemic effects of severe COVID-19 are largely mediated through the immune response to SARS-CoV-2 infection and subsequent inflammatory response. Viral infection stimulates macrophages to overproduce pro-inflammatory cytokines, including IL-6, leading to the 'cytokine storm' that results in systemic inflammatory response syndrome (*Hu et al., 2021*). This heightened inflammatory state affects multiple organs, partly through effects on endothelial cells, which can be directly injured in response to pro-inflammatory cytokines and produce a procoagulant state leading to thrombosis (*Fard et al., 2021*). Improved understanding of the cellular and molecular mechanisms that drive severe COVID-19 and lead to damage in specific organs, as well as the development of long-COVID, requires a multi-organ approach.

We have previously shown that multi-organ, single-cell transcriptome-based approaches can yield significant insights into organ biology and cross-organ signaling (*Tabula Muris Consortium et al., 2018*; *Jones et al., 2022*). In addition, several other studies have recently applied a single cell-based approach to autopsy samples from patients with severe COVID-19. These studies have yielded significant insights into how severe COVID-19 affects the lung (*Delorey et al., 2021*; *Melms et al., 2021*) and the brain (*Yang et al., 2021*), but have not described in detail the systemic and cross-organ effects of severe COVID-19.

Here, we report a COVID-19 single-nuclei RNA-sequencing (snRNA-seq) atlas comprising six organs and approximately 86,000 cells. We showed that transcriptional changes in severe COVID-19 infections were not restricted to the lung, the most severely affected organ upon SARS-CoV-2 infection, but to multiple organs, such as the liver and heart. In addition, we found significant changes in the transcriptional profiles of multiple cell types and identified a subset of recurrent molecular pathways commonly up-regulated in multiple cell types across organs. The COVID Tissue Atlas (CTA) represents a comprehensive resource to investigate the transcriptional changes resulting from COVID-19 in different tissues. Moreover, the scope of the CTA dataset enabled us to identify systemic transcriptional signatures that we would have missed by focusing on an individual organ. We anticipate our analysis and the CTA to be of significant value for future research, including identifying molecular targets for drug development and therapeutic applications.

## Methods
### Sample collection

Organs from post-mortem control individuals and patients with COVID-19 were obtained at the time of autopsy from the University of California, San Francisco Medical Center, and the Saarland University Hospital Institute for Neuropathology. In both institutions, autopsies are exempt from local Institutional Review Board and Committee on Human Research oversight. All autopsies performed have informed consent from legal next of kin for use of tissues for research purposes. *Supplementary file 1* presents all group characteristics.

### Tissue processing

During the autopsy, tissue samples were stored in ice-cold Wisconsin solution for transportation, then immediately processed as follows: tissues were rinsed twice with ice-cold PBS, then wiped off. Next, tissues were pre-cut into 1–2 mm$^3$ cubes, flash-frozen in dry ice, and then stored at –80°C for single-nuclei extraction and total RNA extraction.

## COVID-19 testing

COVID-19 testing was performed on patients according to the testing procedure of host hospitals. For sample testing, total RNA was extracted using a hybrid TRIzol (Life Technologies #15596026) and RNeasy Mini kit (QIAGEN #74104) protocol (*Wolock et al., 2019*; *Rodriguez-Lanetty et al., 2006*). RT-qPCR test for SARS-CoV2 mRNA detection was performed starting from 100 ng of total RNA using a one-Step RT-qPCR enzyme mix (QuantaBio, 94134-500), with primers and probes specific for the SARS-CoV-2 Nucleocapsid N1 and N2 genes, and for human gene ribonuclease PP30 which was used as an internal control (Integrated DNA Technologies, 10006713). The absolute number of transcripts was calculated using a standard curve generated with a positive control for the SARS-CoV2 Nucleocapsid sequence (Integrated DNA Technologies, 10006625).

## Nuclei dissociation

The protocol for nuclei isolation was performed in a BSL2+ biosafety cabinet for the lung and in a BSL2 biosafety cabinet for all other organs wearing personal protective equipment. We carried out all procedures on ice or at 4°C. Single nuclei were generated from around 50 mg of flash-frozen tissues using the Singulator machine (S2Genomics, Livermore, CA, USA), following the manufacturer's recommendations. The extended protocol was used for the ileum and colon, and the regular protocol was used for all other organs. After isolation, nuclei preparations were cleaned as follows: nuclei were centrifuged at 500 × *g* for 5 min and resuspended in 2 ml of cold Storage Buffer (S2Genomics), then centrifuged again at 500 × *g* for 5 min, resuspended in 2 ml of Storage Buffer, and filtered through a 40 µm Flowmi Tip Strainer filter. After centrifugation, nuclei were resuspended in 50–500 µl of Storage Buffer supplemented with 1 U/µl of RNAse inhibitor (Sigma-Aldrich, cat: 3335402001) and counted using a LUNA-FL Dual Fluorescence Cell Counter (Logo Biosystems, Anyang-si, South Korea).

## 10x Genomics protocol

For droplet-based snRNA-seq, libraries were prepared using the Chromium Next GEM Single Cell 3′ v.3.1 according to the manufacturer's protocol (10x Genomics), targeting 10,000 nuclei per sample after counting with a TC20 Automated Cell Counter (Bio-Rad). We performed 12 cycles for cDNA amplification for all of the samples. To generate the final dual or single indexed 10X libraries, 13 cycles were performed.

## Library pooling and quality control

After library preparation, individual libraries were quality checked on an Agilent 4200 Tapestation using D5000 screen tape. These libraries were pooled equal molar into a total of seven pools ranging from 4 to 15 nM final concentration and quality checked again on an Agilent 4200 Tapestation using a D5000 screen tape, followed by qPCR on a Bio-Rad CFX96 RT PCR thermal cycler using the KAPA library quantification kit (# KK4923).

## Sequencing

Individual pools of 10x 3′ gene expression libraries were sequenced on Illumina's Nextseq 2000 P3, Novaseq S2 and/or NovaSeq S4 flow cells with a targeted sequencing read depth of 20,000 reads per cell. Sequencing parameters were as follows: (1) for dual indexed libraries: Read 1=28 cycles, Index 1=10 cycles, Index 2=10 cycles, Read 2=90 cycles; (2) for single indexed libraries: Read 1=28 cycles, Index 1=8 cycles, Index 2=0 cycles, Read 2=91 cycles.

## Alignment

Sequences were de-multiplexed using bcl2fastq version 2.20.0.4.22. Reads were aligned to an extended Gencode Reference 30 (GRCh38) genome containing SARS-CoV2 genes (kindly provided by Aviv Regev and Carly Ziegler) using CellRanger version 5.0.1, available from 10x Genomics, with default parameters.

## snRNA-seq quality control

The count matrices generated by CellRanger were pre-processed by removing contamination of ambient RNA. We noticed high levels of contamination in single-nuclei data, which has been reported before (*Yang et al., 2020*), and applied Cellbender version 0.1 (*Fleming et al., 2019*) to generate

decontaminated count matrices (FDR = 0.01 and default parameters). For quality control, pre-processing, and clustering, we used Scanpy (*Wolf et al., 2018*). We applied quality control filters directly on the count matrices generated by Cellbender. The minimum number of counts per cell we applied as a cut-off varied depending on the sample and ranged between 300 and 800 counts per cell. We observed high mitochondrial content in some of the samples and filtered out cells that exceeded the cut-off threshold (10–20% depending on the sample). We also applied Scrublet for automated identification of potential doublets (*Wolock et al., 2019*).

### Data clustering

For each organ, we first integrated the samples from different donors into a harmonized UMAP embedding using scVI (*Lopez et al., 2018*) release 0.11.0. For training the scVI's variational auto-encoder neural network, we used default parameters except for *n_latent = 64* and *n_layers = 2*. We allowed each gene to have its own variance parameter by setting *dispersion="gene"*. We then used the UMAP algorithm to visualize the resulting embedding in two dimensions. All UMAPs for each organ shown in the manuscript were generated in the same way. The UMAPs generated using scVI's latent space showed minimal batch effect and allowed for the identification of cell populations based on known markers for each organ. For each organ, we first verified that individual clusters expressed known gene markers for the expected cell types. Some clusters, however, co-expressed multiple mutually exclusive markers, an indication of ambient RNA contamination, so we labeled these cells as doublets. Clusters that either expressed gene markers for multiple cell types (doublets) or did not express any markers for the cell types expected in the organ (unidentifiable cells) were systematically removed from the dataset. Finally, for each organ, we generated h5ad files with the cell-type annotations and the harmonized UMAP.

### Cell-type annotation

We used the batch-corrected UMAPs for cell-type annotation. In brief, tissue experts at either UCSF or Stanford (from research labs focused on specific human tissues) analyzed the expression of cell type-specific markers and assigned identities to the clusters. Confident annotations for some clusters, however, were not possible due to high levels of RNA contamination or low expression of marker genes. We therefore only considered clusters for which a cell-type identity was clearly defined. The second round of quality control was applied based on feedback from tissue experts and their annotations. We increased the cut-off values for mitochondrial genes and filtered out putative doublets (cells co-expressing gene markers for mutually exclusive cell types). After the second round of review with the tissue experts, we finalized the cell-type annotations for all organs and used them for all downstream analyses. We use the cell-type label annotations as ground truth for differential expression (DE) analysis, pathway enrichment, and ligand-receptor enrichment analysis (see Signaling interactions between cell types).

### Integration with external datasets

To annotate cell types in the kidney, we integrated our COVID-19 samples with a single-nuclei atlas of the kidney (*Muto et al., 2021*). We applied scANVI (*Xu et al., 2021*) for integration and label transfer. We confirmed that cell types from COVID-19 donors integrated well with the kidney atlas by inspection of cell type-specific markers (*Figure 1—figure supplement 2C-E*). We used the integrated kidney object to compute DE genes and gene pathway enrichment. Additionally, to increase the statistical significance of the identified DE genes, we integrated the COVID-19 and healthy lung single-nuclei samples with the lung data from the Tabula Sapiens dataset (*Consortium and Quake, 2021*). This integration allowed us to increase the number of healthy cells in endothelial cells and macrophages for which we had not enough large populations in our healthy single-nuclei samples. We used scVI to integrate samples from the CTA and Tabula Sapiens and verified that cell types independently identified on each dataset clustered together in the harmonized embedding.

For the sub-clustering and analysis of lung epithelial cells, we independently integrated the CTA lung samples with the COVID-19 lung atlas published by the Broad Institute (*Delorey et al., 2021*) and with the lung dataset (*Kathiriya et al., 2022*). For each data source, we considered only epithelial cells (basal, AT1, and AT2) and performed integration using Seurat 3 (*Stuart et al., 2019*) (correcting batch effect by donor). We kept the original annotations from each dataset to perform comparisons.

We set the default assay parameter to 'RNA' within the integrated dataset to compute the top 10 DE genes. To investigate the transcriptomic differences and similarities between (*Kathiriya et al., 2022*) and the CTA dataset, we generated a hierarchical clustering heatmap by downsampling the datasets to 500 cells per population, using the top 20 genes in the signature gene sets developed in the control dataset. Heatmaps were generated using the R package *pheatmap* v1.0.12 with the clustering algorithm set to *ward.D2*.

## Differential gene expression

To identify DE genes between healthy and COVID-19 samples, we used a negative-binomial model using the *zlm* method as implemented by the MAST R package v1.20 (*Finak et al., 2015*). Following standard practices in single-cell DE, we corrected for the number of detected genes as a potential confounding variable (*Finak et al., 2015*). Finally, to correct the p-values for multiple testing, we applied Bonferroni correction and defined significant DE using an adjusted p-value cut-off of 0.05 and a minimum absolute log2 fold-change (log2-FC) of 1.

## Gene set enrichment analysis

To identify gene sets enriched in COVID-19 donors, we selected the top DE genes for each cell type (COVID-19 vs healthy) and used them as input for pathfindR (*Ulgen et al., 2019*), a gene set enrichment algorithm that includes the fold-change along with potential interactions using a protein-protein interaction network. For selecting significant DE genes, we applied a threshold of log2-FC>*abs*(1) and adjusted p-value <0.001.

We used four different pathway databases as references for our analysis to be comprehensive, KEGG, Reactome, GO, and BioCarta. We then manually curated the enriched pathways, discussed them with tissue experts, and cross-validated them with existing literature to identify the signatures enriched in COVID-19 donors for each cell type and organ. We only considered enriched pathways with a p-value <0.001.

## Coordination in transcriptional responses

To identify transcriptional coordination in COVID-19 samples, we developed a custom analysis method to quantify shared responses across organs. First, we examined the set of genes that appear DE (adjusted p-value <0.001 and log2-FC >*abs*(1)) in at least two-thirds of the organs. Some cell types appear in all organs, whereas some only appear in two or three. We, therefore, applied the coordination analysis only for cell types that appear in at least three organs (macrophages, fibroblasts, and endothelial cells).

We calculated a custom coordination score for each cell type, which was defined as follows. For a pair of organs, we took the set of shared DE genes common to both organs and computed the sign of change for each gene in each organ (i.e., positive/negative for up/down-regulation, respectively). For genes with the same sign in both organs, we assigned a value of 1; genes with opposing signs were assigned a value of 0. The coordination score for the pair of organs was then defined as the average value across shared DE genes (i.e., the sum of values divided by the number of genes). Thus, a coordination score of 1 indicates that all shared DE genes are jointly up- or down-regulated (i.e., perfect coordination). In contrast, a score of 0 indicates that they are oppositely up- or down-regulated (i.e., perfect anti-coordination). For each cell type and for each pair of organs, we thus computed the coordination score.

As a negative control, we repeated this analysis with a computationally shuffled dataset. Here, for each pair of organs for a particular cell type, we held the log2-FC values per gene in one organ fixed and randomly shuffled the log2-FC values per gene in the second organ. We reasoned that this shuffled dataset should possess near-zero coordination (i.e., a score of 0.5), with some small random deviation due to the finite size of the shared gene list. We generated N=1000 computationally shuffled datasets for each pair of organs and calculated the resulting coordination scores for each instance, producing a distribution of coordination scores as a negative control. We then averaged the results and retained the mean and standard error, to be compared with the coordination scores from the actual data. We then used the shared responses as input for pathway enrichment (see above), considering only the cell types that showed significant coordination compared to the random control (macrophages and endothelial cells).

### Signaling interactions between cell types

We applied CellPhoneDB (*Efremova et al., 2020*) and identified significant pairs of ligands and receptors between macrophages and endothelial cells in COVID-19 tissues (adjusted p-value <0.05). We first identified the significant ligand-receptor interactions in healthy and COVID-19 samples independently and considered only those that were enriched in COVID-19 but not in healthy samples.

## Results

### The CTA

We collected data from 20 different autopsies (17 males, 3 females) with an age range between 40 and 89 years (median age = 68 years), of which 15 tested positive for COVID-19 (*Figure 1A*). The average time at which samples were collected was 63 hr post-mortem. Ethnicities were distributed as Hispanic (n=5), African American (n=2), Asian (n=1), and White (n=12). For COVID-19-positive autopsies, the average positive test time before death was 20 days; however, not all donors died due to COVID-19 complications (*Supplementary file 1*). We optimized single-nuclei RNA extraction and sequencing from frozen tissue for Biosafety Level 2 work. All samples were sequenced at the Chan Zuckerberg Biohub using 10x Genomics protocols. After quality control, 85,376 cells (60,946 cells from COVID-19 samples and 24,430 cells from healthy donors) were deemed high quality and used to form the CTA (*Figure 1B and C*). snRNA-seq is prone to high levels of ambient RNA contamination, which we corrected by applying an established correction algorithm (*Fleming et al., 2019*) along with filtering of doublets (Methods). The total numbers of single cells for each organ were as follows: heart (6092 healthy; 13,999 COVID-19), lung (9684 healthy; 11,790 COVID-19), liver (6768 healthy; 8889 COVID-19), prostate (1886 healthy; 8986 COVID-19), kidney (4060 COVID-19), and testis (13,222 COVID-19) (*Figure 1D and E*). Additionally, small intestine, colon, and uninfected control kidney specimens were processed but did not yield sufficient high-quality nuclei for inclusion. We were not able to collect uninfected testis tissue.

We applied dimensionality reduction (PCA) and Leiden clustering for each organ while correcting batch effects across donors using scVI (*Lopez et al., 2018*). Finally, we visualized the resulting clustering using UMAP (*McInnes et al., 2018*). For each organ, we identified cell populations using the batch-corrected UMAP embedding by tissue experts based on the expression of known gene markers (Methods). We identified most major cell types in each organ. We verified that clusters with the same cell identity included healthy and COVID-19 cells, indicating that batch effects were removed (*Figure 1—figure supplements 1 and 2*). Additionally, we verified that our single-nuclei data was statistically comparable to whole-cell sequencing regarding the number of UMIs and detected genes (*Figure 1—figure supplement 3*).

Measurements of SARS-CoV-2 mRNA by RT-qPCR showed high to moderate expression in the lung samples from COVID-19 donors (*Figure 1—figure supplement 4A*). While some of the COVID-19 associated genes such as ACE2, TMPRSS2, and NRP1 were expressed in multiple organs (*Figure 1—figure supplement 4C*), we did not detect significant viral mRNA load by RT-qPCR in the other organs processed (*Figure 1—figure supplement 4A*). The low detection rate of viral mRNA could be attributed to the prolonged periods between initial infection and sample collection for some donors (*Deinhardt-Emmer et al., 2021*; *Supplementary file 1*). Due to the balanced representation of healthy and COVID-19 donors for lung, heart, and liver, we decided to focus our downstream analysis on understanding the transcriptional responses of cell types in these organs. For the kidney, we integrated our data with a healthy single-nuclei atlas reference (*Muto et al., 2021*) and made the integrated object available (Data Availability). The results for DE analysis between COVID-19 and healthy samples for lung, heart, liver, kidney, and prostate are available as part of the CTA data release (*Supplementary files 2 and 3*). Finally, our testis data, including only COVID-19 samples, is fully annotated and publicly available as part of the CTA release.

### Cell-type population changes in the COVID-19 lung

The CTA lung dataset comprised 21,474 cells, of which 11,790 were collected from COVID-19 autopsies. After quality control and clustering (Methods), we identified 10 distinct cell types, including primary epithelial and immune cells (*Figure 2—figure supplement 1A, B*). Several lung single-nuclei and single-cell efforts have been published throughout the COVID-19 pandemic (*Delorey et al., 2021*;

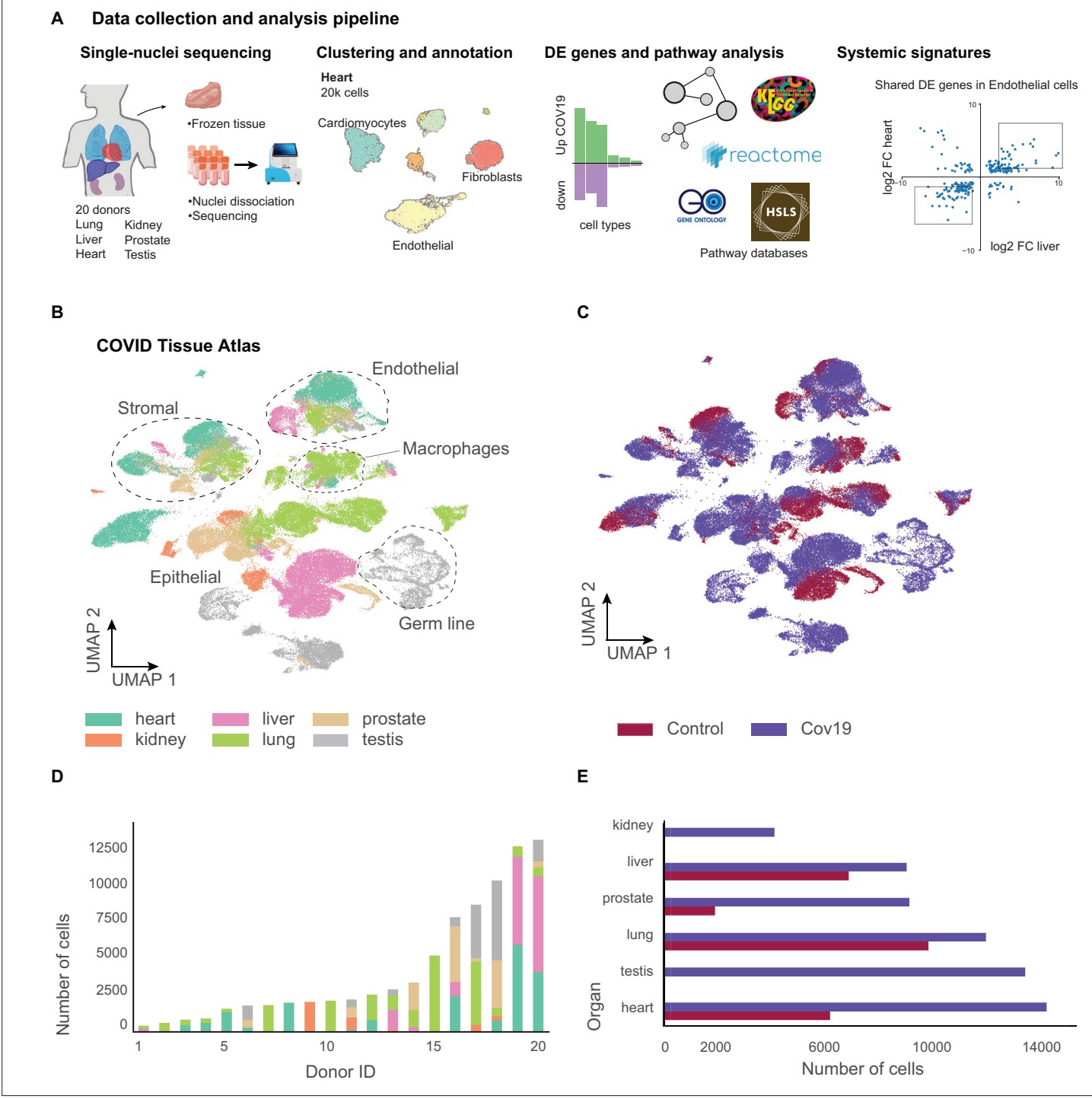

**Figure 1.** A human single-cell atlas enables the identification of systemic responses to COVID-19. (**A**) Tissue samples were collected from different organs and frozen, then dissociated into single nuclei. Libraries for single-nuclei RNA-sequencing (snRNA-seq) were prepared using 10x Genomics Chromium Next GEM Single Cell 3' v.3.1kit, followed by sequencing on various Illumina platforms. After quality control and clustering, cell types for each organ were annotated by experts using literature gene markers. Differential gene expression and pathway enrichment analysis were performed between COVID-19 and healthy samples for all cell types. Finally, global transcriptional signatures were identified via a cross-organ analysis of differential expression. (**B**) The COVID tissue atlas comprises approximately 85,000 cells from six different organs. (**C**) Cells in the COVID tissue atlas cluster by cell identity rather than disease status. (**D**) Number of cells per donor grouped by the organ of origin. (**E**) Number of cells per organ grouped by COVID-19 status.

The online version of this article includes the following figure supplement(s) for figure 1:

*Figure 1 continued on next page*

**Figure supplement 1.** Clustering, batch correction, and annotation of single-nuclei samples.

**Figure supplement 2.** Clustering and annotation of additional organs.

**Figure supplement 3.** Comparison between whole-cell and single-nuclei RNA-seq.

**Figure supplement 4.** Detection of viral mRNA and expression of COVID-19 associated genes.

*Xu et al., 2020*; *Hasan et al., 2021*; *Melms et al., 2021*). To assess the quality and scope of the CTA, we compared our data to the comprehensive lung atlas generated by the Broad Institute (*Delorey et al., 2021*). We applied anchor-based integration (*Stuart et al., 2019*) to the lung samples from both datasets by including autopsies from the Broad atlas as additional donors in the CTA (Methods). After integration, the harmonized UMAP embedding showed that all major lung cell types integrated well across datasets (*Figure 2A*), with cells from the CTA and the Broad atlas contributing to most clusters (*Figure 2B*). The alignment between datasets showed that the CTA captured the expected diversity of cell types in the COVID-19 lung and that the gene expression profiles were similar for the same cell types across datasets.

We next focused on the effects that COVID-19 has on the different lung cell populations. In particular, significant epithelial cell damage resulting from COVID-19 is manifested as loss of alveolar type 1 (AT1) and alveolar type 2 (AT2) cells (*Melms et al., 2021*; *Delorey et al., 2021*). To investigate the changes in lung epithelial cells in COVID-19 autopsies in detail, we subset and re-clustered the AT1, AT2, and basal cells to obtain a new UMAP embedding (*Figure 2C*). All three cell types included healthy and COVID-19 cells (*Figure 2—figure supplement 1C*) and expressed the corresponding canonical gene markers (*Figure 2—figure supplement 1D*). Consistent with previous studies (*Delorey et al., 2021*; *Melms et al., 2021*), we identified loss of AT1 and AT2 cells in COVID-19 lungs relative to healthy controls (*Figure 2D*, *Figure 2—figure supplement 1A*), along with a significant expansion of basal cells (*Figure 2D*, *Figure 2—figure supplement 1A*).

COVID-19 is known to cause severe alveolar damage and loss of epithelial cells. During alveolar injury, AT2 cells are known to serve as progenitors to generate new epithelial cells. More recently, an additional cellular pathway has been identified where AT2-derived basaloid cells generate AT1 cells, termed alveolar basal intermediates. These cells were first identified in single-cell studies as KRT17+/KRT5- basaloid cells in human idiopathic pulmonary fibrosis (IPF) lungs (*Habermann et al., 2020*) and independently shown by in vitro studies to arise from AT2 cells (*Kathiriya et al., 2022*). To find if the increase in basal cells in COVID-19 lungs could be explained by trans-differentiation from AT2 cells, we integrated the CTA lung epithelial cells with an sc-RNA-seq dataset of lungs with IPF comprising two populations of transitional cells: KRT17+/KRT5- ABI cells, and transitional AT2 cells with a similar transcriptomic profile to AT1 cells (*Habermann et al., 2020*). After applying anchor-based integration (Methods), the harmonized UMAP showed that epithelial cells from both datasets generally integrated well (*Figure 2E*).

Interestingly, while most of the transitional cells from *Habermann et al., 2020*, mapped to the AT1 cluster, likely corresponding to transitional AT2s, a smaller fraction mapped to a specific cluster between the AT2 and basal cell populations from the CTA (*Figure 2E*, right). To verify if this cluster from the CTA indeed corresponds to ABIs, we looked at genes up-regulated in ABIs (*Kathiriya et al., 2022*). We found that the ABI gene expression profile was predominantly expressed within basal cells in CTA (*Figure 2F*). Thus, our data suggest that ABIs (*Kathiriya et al., 2022*) are present in the COVID-19 lung, and their trans-differentiation from endogenous AT2s could account for the gain of abnormal basal cells in the alveoli, which are lost in COVID-19 lungs.

Alternatively, the Broad atlas identified a pre-AT1 transitional cell state (PATS) population in COVID-19 lungs (*Delorey et al., 2021*) that bears similarities to what was previously described as ABIs/transitional AT2s/aberrant basaloid cells from IPF lungs (*Adams et al., 2020*; *Habermann et al., 2020*; *Kathiriya et al., 2022*). We jointly analyzed the lung epithelial cells from the CTA and the Broad atlas to determine if the PATS population was present in our data. We used anchor-based integration (*Stuart et al., 2019*) and obtained a harmonized UMAP embedding which recapitulated the three populations across datasets (*Figure 2G*). The PATS population mostly overlapped with the principal AT1 cluster (*Figure 2G*, right), but no specific cluster from the CTA mapped directly to the PATS cells. This analysis indicates that the PATS population (*Delorey et al., 2021*) is likely attributed

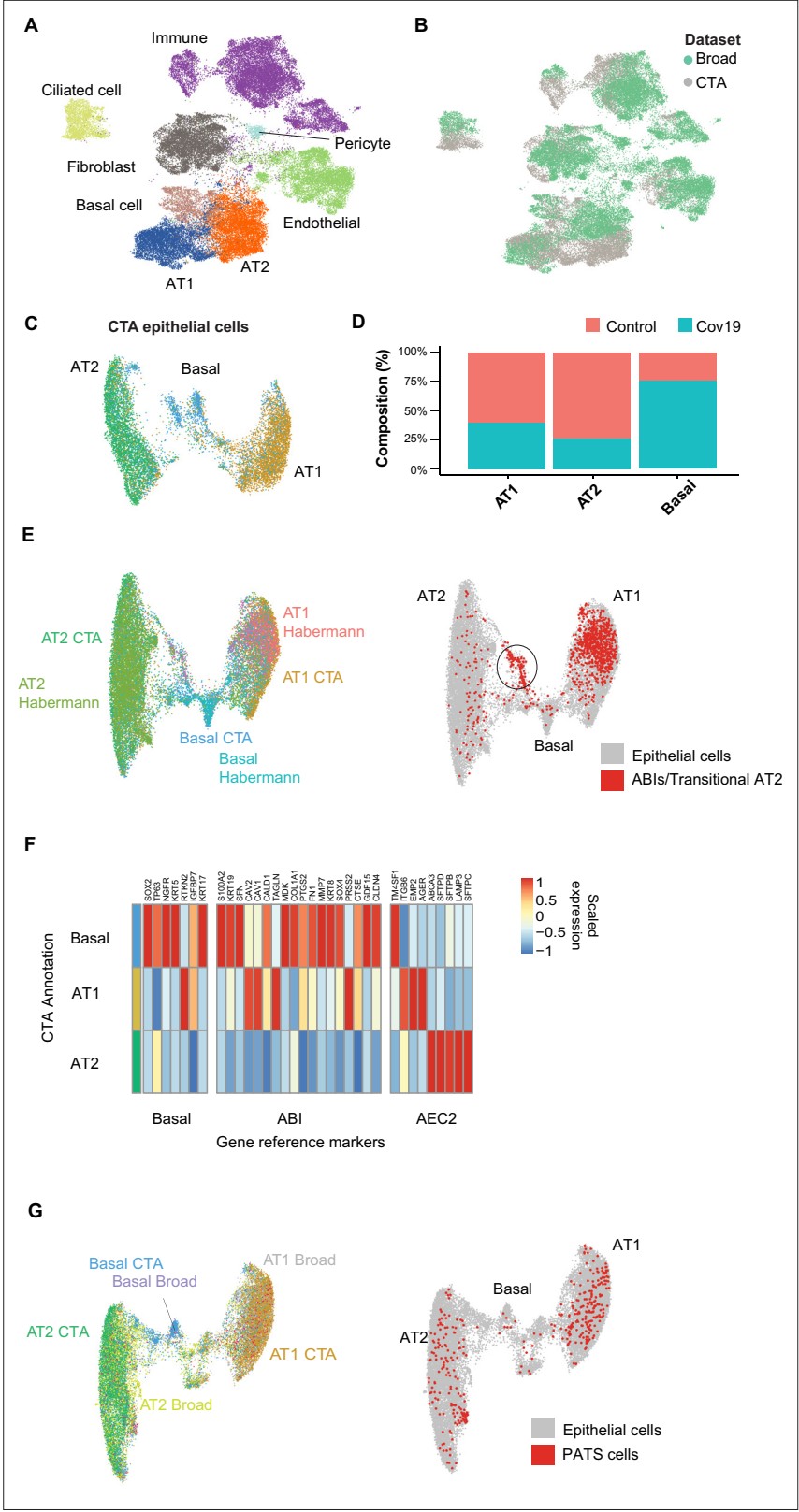

**Figure 2.** Cell type composition changes in the COVID-19 lung. (**A**) Integration of the COVID Tissue Atlas (CTA) lung with the lung COVID atlas by the Broad Institute. A harmonized UMAP shows that cells from both datasets integrate by their corresponding cell-type annotation. (**B**) Integration of two lung COVID atlas colored by the dataset of origin. (**C**) Sub-clustering and UMAP projection of the CTA lung epithelial cells (AT1, AT2, and basal

*Figure 2 continued on next page*

*Figure 2 continued*

cells). (**D**) Relative cell composition in epithelial lung tissue from control and COVID-19 autopsies (CTA data only). (**E**) Integration of CTA epithelial cells and epithelial cells from *Habermann et al., 2020* (AT1, AT2, basal cells, and transitional ABIs/AT2 populations). ABIs/transitional AT2 from *Habermann et al., 2020*, are shown in red (right). (**F**) Heatmap of scaled gene expression of marker genes for three different reference cell populations (x-axis) compared to the CTA lung cell types (y-axis). (**G**) Joint embedding of CTA and *Delorey et al., 2021* (AT1, AT2, basal, and pre-transitional cell state [PATS] cells). The PATS cells identified by *Delorey et al., 2021*, are shown in red in the joint UMAP (right).

The online version of this article includes the following figure supplement(s) for figure 2:

**Figure supplement 1.** Cell-type composition of the COVID-19 lung.

to patient-specific cellular heterogeneity (or sequencing method differences) and, therefore, was not detected in the CTA donors. Together, our results contribute to our understanding of the multiple regenerative strategies involved in re-establishing alveolar epithelial homeostasis in response to COVID-19 (*Delorey et al., 2021*).

## Insulin signaling dysregulation in the liver

Across all six cell types identified in the liver (*Figure 3A*), hepatocytes comprised around 60% of cells in the healthy samples and more than 80% in the COVID-19 samples (*Figure 3B*). However, we observed an inverse trend for endothelial cells, where approximately 20% of the cells from healthy samples were annotated as endothelial as opposed to less than 10% in COVID-19 donors, which may reflect recently reported endotheliopathy in COVID-19 livers (*McConnell et al., 2021*). COVID-19 livers also contained lower proportions of most immune cell populations than controls (*Figure 3B*).

To identify DE genes for each cell type in the liver, we applied a negative-binomial model implemented in MAST (*Finak et al., 2015*) that corrects for differences in sequencing depth across samples. Across all cell types, hepatocytes, endothelial cells, and macrophages showed the largest number of DE genes in COVID-19 donors (more than 200 up-regulated genes with an average log2-FC >2 and adjusted-p <1e-6; *Figure 3C*). In contrast, fibroblasts, intrahepatic cholangiocytes, and natural killer cells showed only a fraction of DE genes in comparison (fewer than 50 up-regulated genes; *Figure 3C*). Samples from COVID-19 livers generally comprised lower counts per cell (*Figure 3—figure supplement 1*); while we corrected for this difference when computing DE genes (Methods), we decided to focus on COVID-19 over-expressed genes to minimize potential artifacts in down-regulation resulting from lower sequencing depth.

Next, we applied pathway enrichment using pathfindR, an algorithm that identifies significant sets of genes based on a reference pathway database and a protein-protein interaction network (*Ulgen et al., 2019*). We identified dysregulated signaling pathways in COVID-19 livers using four different reference pathway databases: KEGG (*Kanehisa et al., 2016*), BioCarta (*Nishimura, 2001*), GO (*Mi et al., 2019*), and Reactome (*Griss et al., 2020*). We found known COVID-19-related gene sets in hepatocytes and macrophages ('*Coronavirus disease - COVID-19*' in the KEGG database), including TMPRSS2, EGFR, PLCG2, MAPK14, FOS, JUN, IFNAR1, C5AR1, CFB, C8G, MASP1, FGA, FGB, FGG in addition to multiple ribosomal-related transcripts, p<1e-6 (*Supplementary file 3*). The expression of known COVID-19 genes indicates general agreement between our data and previous studies (*Harrison et al., 2020*).

Several pathways were enriched in multiple cell types in the liver across all four databases, including insulin, HIF-1, Notch, MAPK, and FoxO signaling (*Figure 3D*, *Supplementary file 3*). We found dysregulation in the insulin signaling pathway in hepatocytes, macrophages, and endothelial cells from COVID-19 livers (*Supplementary file 3*; p<1e-6). Specifically, we observed up-regulation of genes involved in insulin response, including INSR, PIK3R1, PIK3CB, GSK3B, PPP1CB, PHKA2, PRKAR1A, SORBS1, CBL, CBLB, ACACA, HK1, PRKAG2, RPS6, RHEB, PTPN1 (*Figure 3E*, *Supplementary file 2*). Patients with type 2 diabetes have worse outcomes with severe COVID-19 infection (*Xie and Al-Aly, 2022*), and clinical studies show aberrant glucose levels in SARS-CoV2 infected patients with type 2 diabetes (*Reiterer et al., 2021*). Thus, our data suggest that dysregulated insulin signaling – especially in hepatocytes, which play a critical role in maintaining glucose homeostasis (*Klover and Mooney, 2004*) – might explain why SARS-CoV2 infected patients with type 2 diabetes

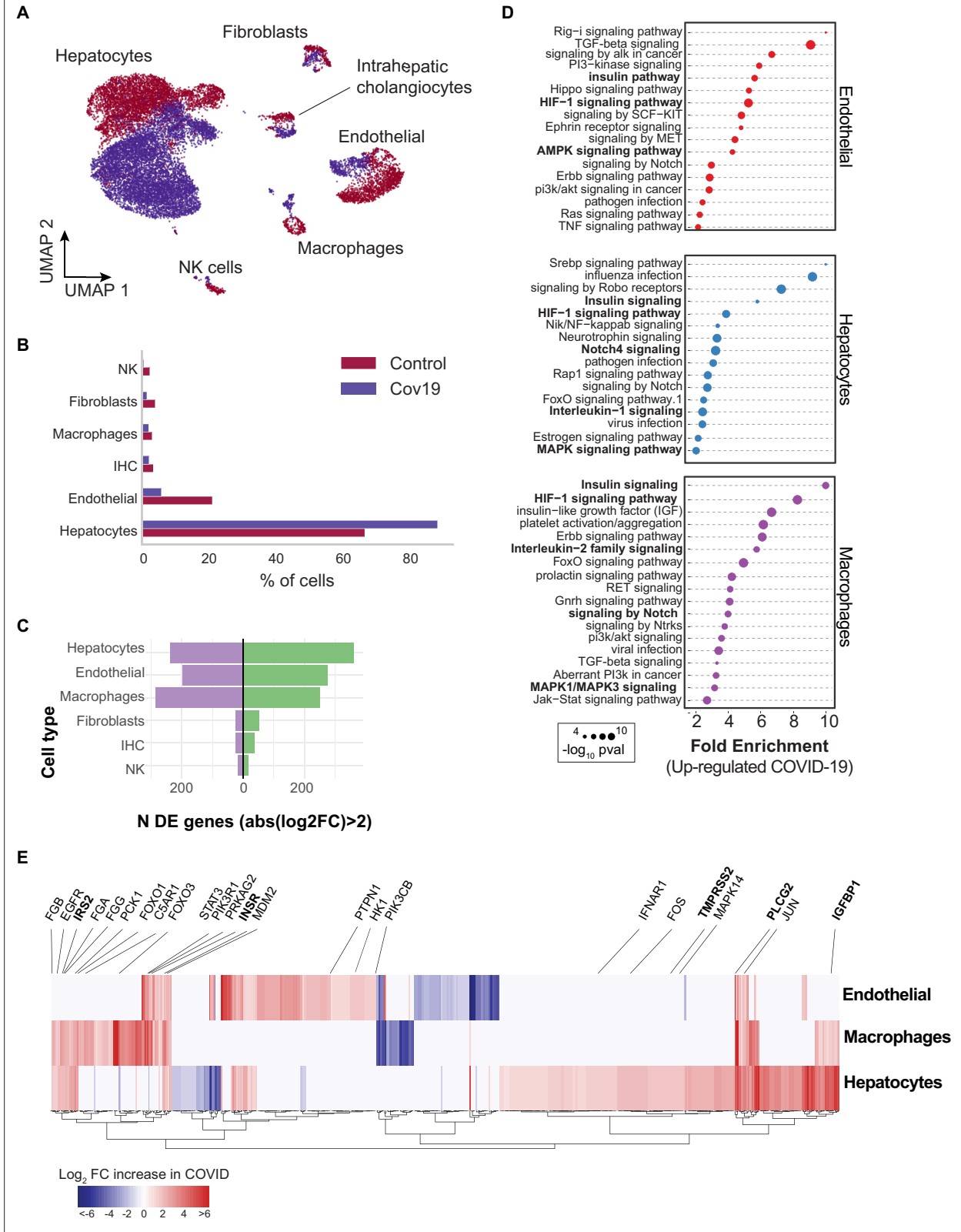

**Figure 3.** Transcriptional changes and dysregulation of cell signaling in the COVID-19 liver. (**A**) UMAP plot showing all cells from liver samples (n=6 donors) colored by COVID-19 status. Cell-type annotations are indicated for each cluster. (**B**) Fraction of cells for each cell type grouped by COVID-19 status. (**C**) The number of differentially expressed (DE) genes found using MAST (**Finak et al., 2015**) (negative-binomial model, correcting for the

*Figure 3 continued on next page*

*Figure 3 continued*

number of detected genes, p<1e-6 and log2 fold change (log2-FC) >2). (**D**) The top enriched signaling pathways found for each cell type based on the DE genes shown in (C). (**E**) Heatmap of log2-FC for the top DE genes. A few relevant genes are highlighted with a text legend.

The online version of this article includes the following figure supplement(s) for figure 3:

**Figure supplement 1.** Sequencing depth in healthy and COVID-19 samples.

**Figure supplement 2.** Endothelial subpopulations in the lung.

**Figure supplement 3.** Endothelial subpopulations in the heart.

have uncontrolled glucose homeostasis and are comorbid (*Mishra and Dey, 2021*) and why COVID-19 infection could lead to the development of type 2 diabetes (*Barrett et al., 2022*).

## Signaling in the heart in response to COVID-19

COVID-19 can lead to cardiac involvement and injury via the following possible mechanisms: (1) indirect injury due to increased cytokines and immune inflammatory response, (2) direct invasion of cardiomyocytes by SARS-CoV-2, and (3) respiratory damage from the virus causing hypoxia leading to oxidative stress and injury to cardiomyocytes (*Tahir et al., 2020*). To understand the transcriptional changes induced by COVID-19 in the heart, we analyzed differential gene expression across cell types and identified the critical signaling pathways dysregulated in response to COVID-19.

Heart samples yielded 20,091 cells after quality control (n=11 donors) (*Figure 4A*). Across the eight cell types identified, the large majority of cells corresponded to endothelial cells (>40% in COVID-19 samples, 13% in healthy samples), cardiomyocytes (25% in COVID-19 samples, 28% in healthy samples), and fibroblasts (15% in COVID-19 samples, 45% in healthy samples) (*Figure 4B*). To further explore the changes in endothelial cells, we independently re-clustered the endothelial cells from the lung and the heart and annotated sub-cell types based on known markers [*Figure 3—figure supplement 2a, b, c* (*Litviňuková et al., 2020*)]. We identified four clusters of endothelial cells in the lung that we classified as vascular, capillary, and aerocytes, based on their expression profiles (*Figure 3—figure supplement 2d*). We then compared the relative composition of endothelial sub-cell types in healthy and COVID-19 samples and noticed an increase in the relative composition of capillary and aerocyte sub-classes (*Figure 3—figure supplement 2f*). Consistent with this, gene ontology (GO) analysis of the DE genes from capillary endothelial cells in the lungs showed that angiogenesis genes are significantly up-regulated (*Figure 3—figure supplement 2h*). Additionally, following a similar analysis in the heart, we found an increase in capillary and arterial endothelial sub-cell types (*Figure 3—figure supplement 3*). Therefore, the higher number of endothelial cells in COVID-19 tissues is correlated with an increasing population of capillary endothelial cells.

We then found significant transcriptional changes in cardiomyocytes, endothelial cells, and macrophages based on the number of DE genes in COVID-19 samples (*Figure 4C*). Considering the top DE genes for each cell type, we then focused on understanding how COVID-19 affects heart cells in terms of gene regulatory pathways. We first confirmed that our results agreed with current gene sets associated with COVID-19 (KEGG: *Coronavirus disease - COVID19* pathway in fibroblasts and macrophages, p<1e-5; Reactome: *Influenza infection* enriched in fibroblasts, p<1e-5; *Supplementary file 3*). In addition, multiple genes and GO pathways related to protein translation and ribosome activity (RNA polymerase II cis-regulatory region sequence-specific DNA binding), along with signaling and transcription factor activity (intracellular signal transduction, transcription cis-regulatory region binding, transcription factor binding), were enriched in multiple COVID-19 heart cell types (*Supplementary file 3*).

Similar to the liver, we observed insulin pathway enrichment in cardiomyocytes from COVID-19 samples (*Figure 4C*). Heart failure is associated with generalized insulin resistance. Moreover, insulin-resistant states such as type 2 diabetes mellitus and obesity increase the risk of heart failure even after adjusting for traditional risk factors (*Riehle and Abel, 2016*). In agreement with our data, other studies found that COVID-19 triggers insulin resistance in patients, causing chronic metabolic disorders that were non-existent before infection (*Govender et al., 2021*). Additionally, we observed significant changes in Notch, Hippo, and MAPK signaling pathways in cardiomyocytes from COVID-19 samples (*Figure 4D*). Conversely, the BMP and TGFβ signaling pathways showed specific down-regulation in endothelial cells from COVID-19 hearts, including down-regulation of BMPR1A, BMPR1B, SMAD6, and BMP6 (*Supplementary file 3*).

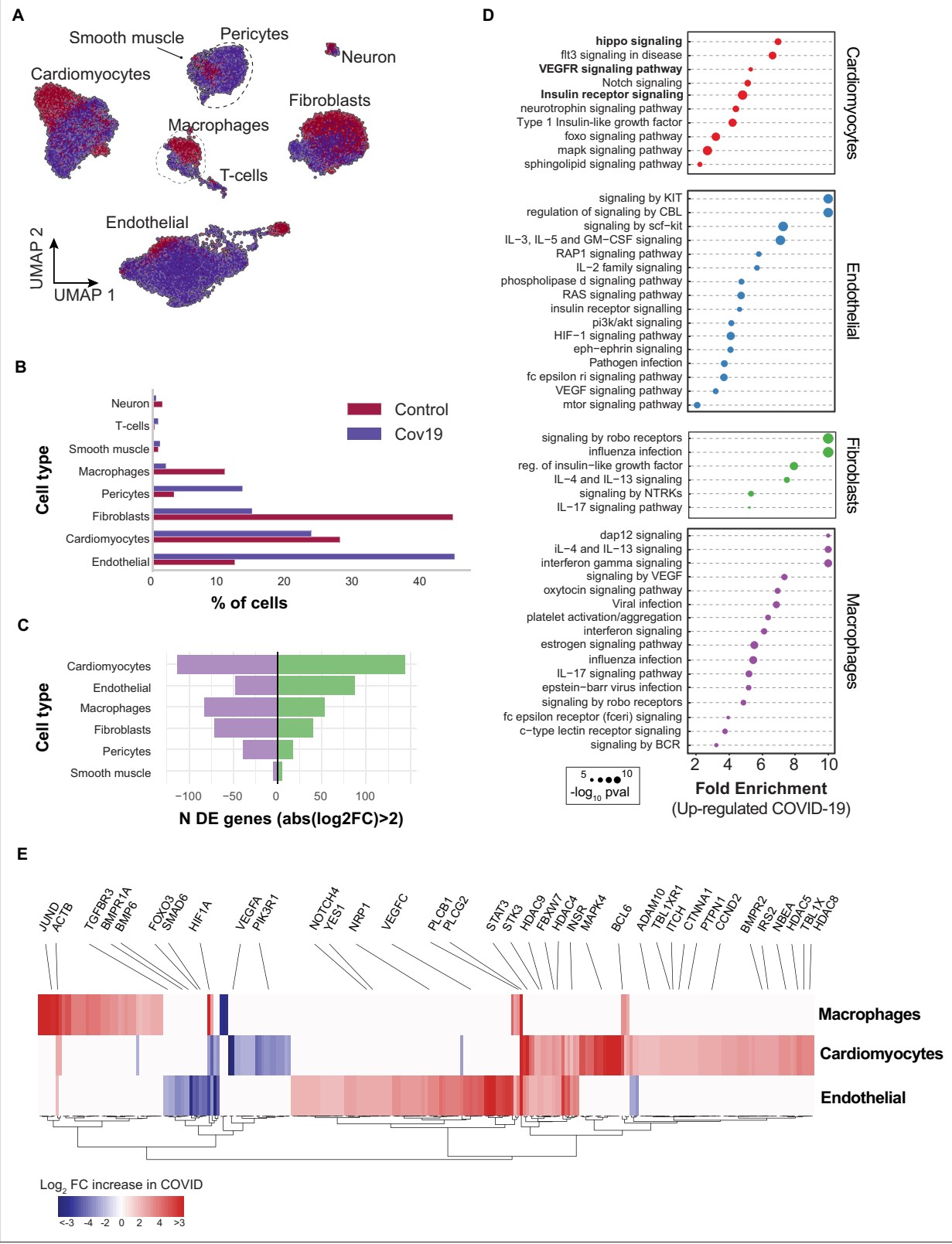

**Figure 4.** Transcriptional changes and dysregulation of cell signaling in the COVID-19 heart. (**A**) UMAP plot showing all cells from heart samples (n=11 donors) colored by COVID-19 status. Cell-type annotations are indicated for each cluster. (**B**) Fraction of cells for each cell type grouped by COVID-19 status. (**C**) The number of differentially expressed genes found using MAST (*Finak et al., 2015*) (negative-binomial model, correcting for the number of detected genes, p<1e-6 and log2 fold change (log2-FC) >2). (**D**) The top signaling pathways found for each cell type using the genes in (C). (**E**) Heatmap of log2-FC for the top differentially expressed genes. A few relevant genes are highlighted with a text legend.

Interestingly, Notch signaling has been proposed as a target to prevent SARS-CoV-2 infection and interfere with the progression of COVID-19 associated heart and lung disease (*Rizzo et al., 2020*). Hippo signaling also appeared as one of the top signaling terms for cardiomyocytes (*Figure 4D*). Recent studies indicate that Hippo signaling is involved in the development of many diseases caused by viruses. Whether virus-induced diseases, specifically COVID-19, can be ameliorated by modulating the Hippo signaling pathway is worth pursuing (*Wang et al., 2019*). Finally, TGFβ signaling is linked to the response of endothelial cells to inflammation in COVID-19 (*Yoshimatsu and Watabe, 2022*). Together, these results build on previously reported evidence to show that multiple signaling pathways in the heart undergo both cell type-specific and systemic changes in response to COVID-19.

## Shared transcriptional responses across organs

The CTA provides a unique opportunity to identify systemic transcriptional responses across organs. As an indication of a systemic response to COVID-19, we found enrichment of the same signaling pathways in multiple cell types and across organs, including HIF-1, insulin, and Notch signaling (*Figures 3D and 4D*). Therefore, we decided to quantify the cross-organ transcriptional changes in COVID-19 autopsies by finding overlapping sets of DE genes and signaling pathways across organs.

In macrophages, we found a significant overlap in DE genes across organs compared to random sampling expectations (*Figure 5A*). Specifically, we found a set of 89 DE genes in COVID-19 macrophages from all three organs, including PLCG2, HIF1A, ACTB, and JUND. There were also many overlapping DE genes in pairs of organs, with macrophages from the liver and lung showing the highest overlap with 124 shared DE genes (*Figure 5A*, *Supplementary file 4*). We performed the same analysis for endothelial cells and also found sets of overlapping DE genes, the highest overlap occurring between endothelial cells from the liver and lung (*Figure 5—figure supplement 1* and *Supplementary file 4*).

To further analyze these data, we defined the shared transcriptional response (STR) for a cell type as the set of genes that show DE in at least three organs from COVID-19 donors (p<1e-4 and log2-FC >1). We restricted the analysis to macrophages, endothelial cells, and stromal cells, which appear in multiple organs, and calculated the correlation between the log-FC values of all genes in the STR across pairs of organs (*Figure 5B–E*). Generally, we saw high correlation coefficients, indicating coordination in the COVID-19 induced STR across organs. For example, among the genes with the highest log-FC across organs, we found HIF1A (in macrophages from liver, lung, and heart; *Figure 5B–C*), JUND (in macrophages from liver and heart; *Figure 5C*), and PLCG2 (in endothelial cells from liver and heart; *Figure 5D*).

To identify the cell types with high coordination in their STR across organs, we defined a coordination score by considering pairs of organs and the fraction of genes in the STR that showed the same direction in DE (upregulated in both or down-regulated in both; *Figure 5F*). Finally, we generated a null distribution for the expected coordination by shuffling the log-FC across genes (*Figure 5F*; Methods) and computed a z-score between the null distribution and the observed coordination for each cell type and pair of organs (*Figure 5G*).

The STR of endothelial cells from COVID-19 samples showed the highest coordination across multiple pairs of organs (*Figure 5G*). These results are consistent with previous studies focused on the effect of COVID-19 on endothelial tissues (*Ruhl et al., 2021*; *Huertas et al., 2020*). We also found significant coordination in macrophages across the liver, lung, and heart (z-score >5; *Figure 5G*). Macrophages from the lung showed lower coordination scores compared to the heart and liver, an indication of lung-specific transcriptional regulation (*Figure 5G* and *Figure 5B* off-diagonal quadrants). In contrast, the STR of fibroblasts and stromal cells showed no significant coordination compared to the randomized control, possibly due to the low number of overlapping DE genes (*Figure 5G* and *Figures 3C and 4C*).

To further validate the SRT across multiple organs, we applied our analysis to the Broad COVID-19 atlas (*Delorey et al., 2021*). The Broad atlas includes lung, liver, and heart samples from COVID-19 autopsies. We first computed DE and log2-fold enrichment between controls and COVID-19 samples for macrophages and endothelial cells on each organ independently. As described above, we found the coordinated DE as the overlapping sets of genes that showed significant differences in all three organs. From 125 significantly expressed genes found in the SRT response of macrophages in the CTA, 65 also appeared in the Broad atlas, representing a 52% overlap across datasets (equivalently,

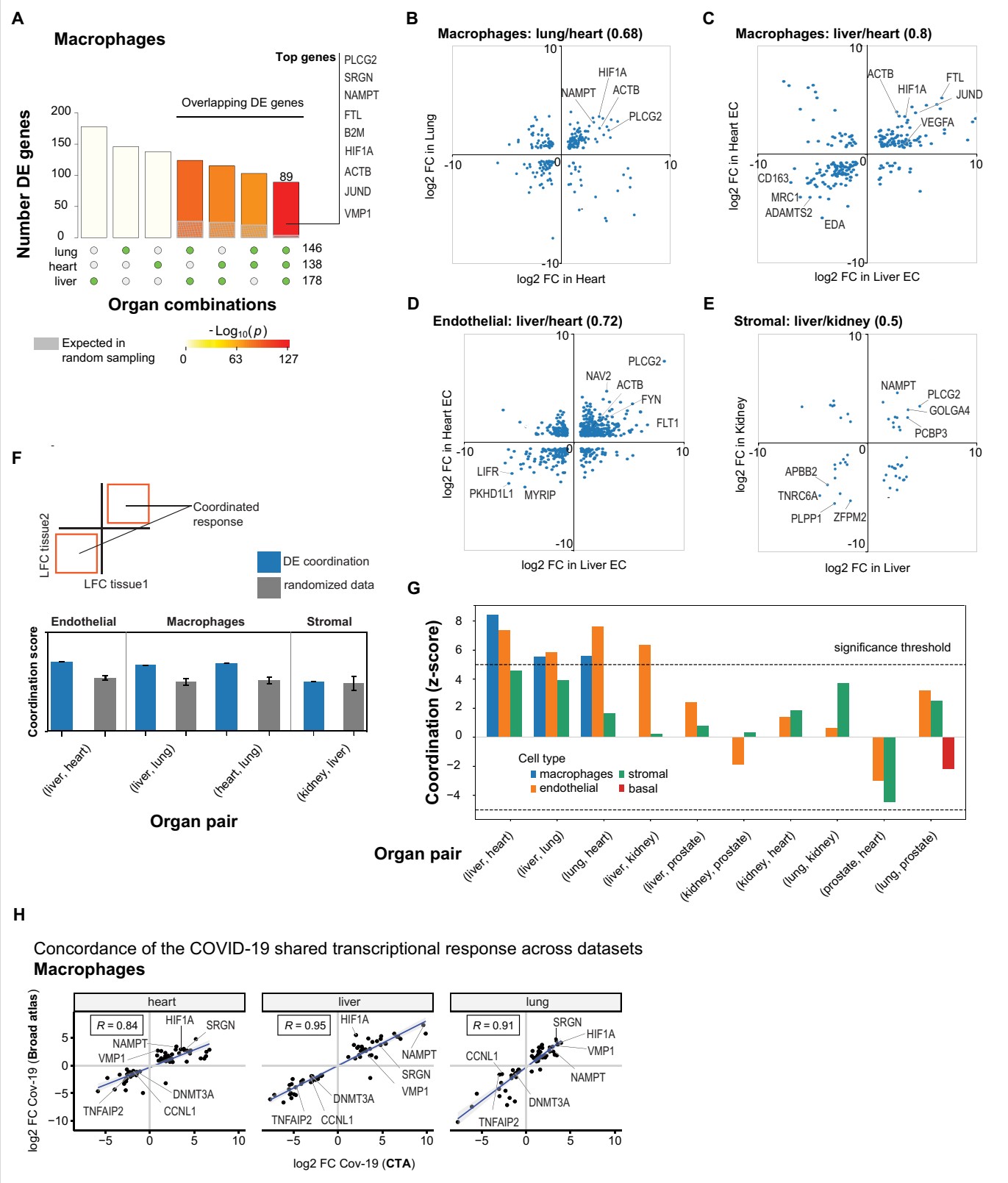

**Figure 5.** A shared transcriptional response in macrophages and endothelial cells across organs. (**A**) Overlap of differentially expressed (DE) genes in COVID-19 macrophages across organs. The gray shaded area indicates the expected overlap for each organ combination (green circles) under a null hypothesis of random sampling (we computed the p-values against this null model). The white bars indicate the number of genes that showed DE in a single organ. The names of the top genes DE in all three organs are shown based on their log2-fold enrichment. (**B**) Scatter plot comparing the log2 fold

*Figure 5 continued on next page*

*Figure 5 continued*

change (log2-FC) for DE genes in COVID-19 macrophages from lung and heart. (**C**) Same as (**B**) but compares DE genes in COVID-19 macrophages from the liver and heart. (**D**) log2-FC for COVID-19 endothelial cells from the liver and heart. (**E**) log2-FC for COVID-19 stromal cells from the liver and kidney. (**F**) A fully coordinated transcriptional signature would imply that all genes lie in the bottom-left and top-right quadrants (red squares). We define the coordination score as the number of DE genes that show the same direction (up-up, down-down) for the two organs, divided by the number of shared DE genes (**A**). Gray bars show the score expectation when sampling DE genes randomly from each organ. (**G**) Coordination scores for different cell types across all pairs of organs. The dotted line indicates a significance threshold of z-score >5 standard deviations compared to the expectation by chance. (**H**) Scatterplot of log2-FC between control and COVID-19 for macrophages from three different organs. Each dot represents a gene; the x-axis shows the log2-FC found using data from the COVID Tissue Atlas (CTA), while the y-axis shows the corresponding log2-FC found using the Broad COVID-19 atlas. The same methods and significance thresholds were applied to both datasets. The scatterplot shows the genes that showed significant differences in both datasets. A few examples with coordinated upregulation across all three organs in both datasets are indicated by name.

The online version of this article includes the following figure supplement(s) for figure 5:

**Figure supplement 1.** Overlapping differentially expressed (DE) genes in endothelial cells across organs.

we found a 31% overlap for the SRT in endothelial cells – *Supplementary file 7*). Finally, we compared the log2-FC magnitudes for the overlapping genes across datasets (*Figure 5H*). The strength of the correlation between log2-FC across independent datasets indicates that the differences we found in the CTA and the coordination of transcriptional responses in multiple organs are robust and likely generalizable in COVID-19. Together, these results indicate that COVID-19 infection induces coordinated transcriptional regulation in macrophages and endothelial cells across multiple organs.

## Systemic transcriptional responses in endothelial cells and macrophages

To investigate the relevance of the COVID-19 STR in macrophages and endothelial cells, we identified enriched pathways considering the genes that showed coordinated DE in at least three organs (diagonal quadrants in *Figure 5B–E*, *Supplementary file 4*). We visualized the results as a matrix of pathways vs. organs, including the top pathway terms (p<1e-3, fold enrichment >3) that appeared in at least two organs for macrophages (*Figure 6A*) and endothelial cells (*Figure 6B*). Multiple signaling pathways were enriched in the STR of macrophages across organs (*Figure 6A*). The HIF-1 pathway showed high fold enrichment across all organs (*Figure 6A*), suggesting a pivotal role of macrophages in the systemic response to oxygen homeostasis in COVID-19. The Notch pathway was also enriched in macrophages from all three organs (*Figure 6A*), confirming that Notch signaling has a crucial role in the systemic response to COVID-19 (*Breikaa and Lilly, 2021*; *Farahani et al., 2022*).

The STR consists of shared DE genes across multiple organs; however, the magnitude of the DE of a given gene, in terms of log-FC and p-value, can vary across organs (*Figure 5B–E*, *Supplementary file 4*). Therefore, when performing pathway analysis, some signaling pathways showed statistically significant enrichment only in subsets of tissues. For example, in macrophages, interleukin-4/13 showed significant enrichment only in the liver and heart and the adherens junction pathway only in the liver and the lung (*Figure 6A*). Similarly, a few gene pathways showed organ-specific enrichment (*Supplementary file 5*), indicating that genes in the SRT, while simultaneously DE across organs, might also modulate some cellular processes in organ-specific ways due to quantitative DE differences.

In the coordinated STR of endothelial cells, we found multiple enriched pathways, including Notch and Ephrin signaling in the lung, liver, and heart (*Figure 6B*). Specifically, several Notch-related genes were upregulated in COVID-19 samples for all three organs, including HDAC9, a selective regulator of Notch, FBXW7, a regulator of angiogenesis through Notch (*Izumi et al., 2012*), and TBLR1, an indirect Notch regulator through degradation (*Perissi et al., 2008*). Additionally, we found enrichment for VEGF signaling in liver and heart (p<1e-3, and lung p<1e-2; *Supplementary file 6*). Interestingly, despite upregulation of the VEGF signaling pathway in multiple organs, some pathway genes showed organ-specific regulation. For example, AKT3 was enriched in the liver and lung, whereas PXN contributed to VEGF signaling enrichment only in the heart and lung (*Supplementary file 6*). A recent study using measurements of growth factors and cytokines in serum identified VEGF-D as the most predictive indicator for the severity of COVID-19 (*Kong et al., 2020*). Similarly, VEGF was proposed as a promising therapeutic target for suppressing inflammation during SARS-CoV-2 infection (*Yin et al., 2020*). Our results indicate that changes in VEGF signaling in COVID-19 donors are not necessarily

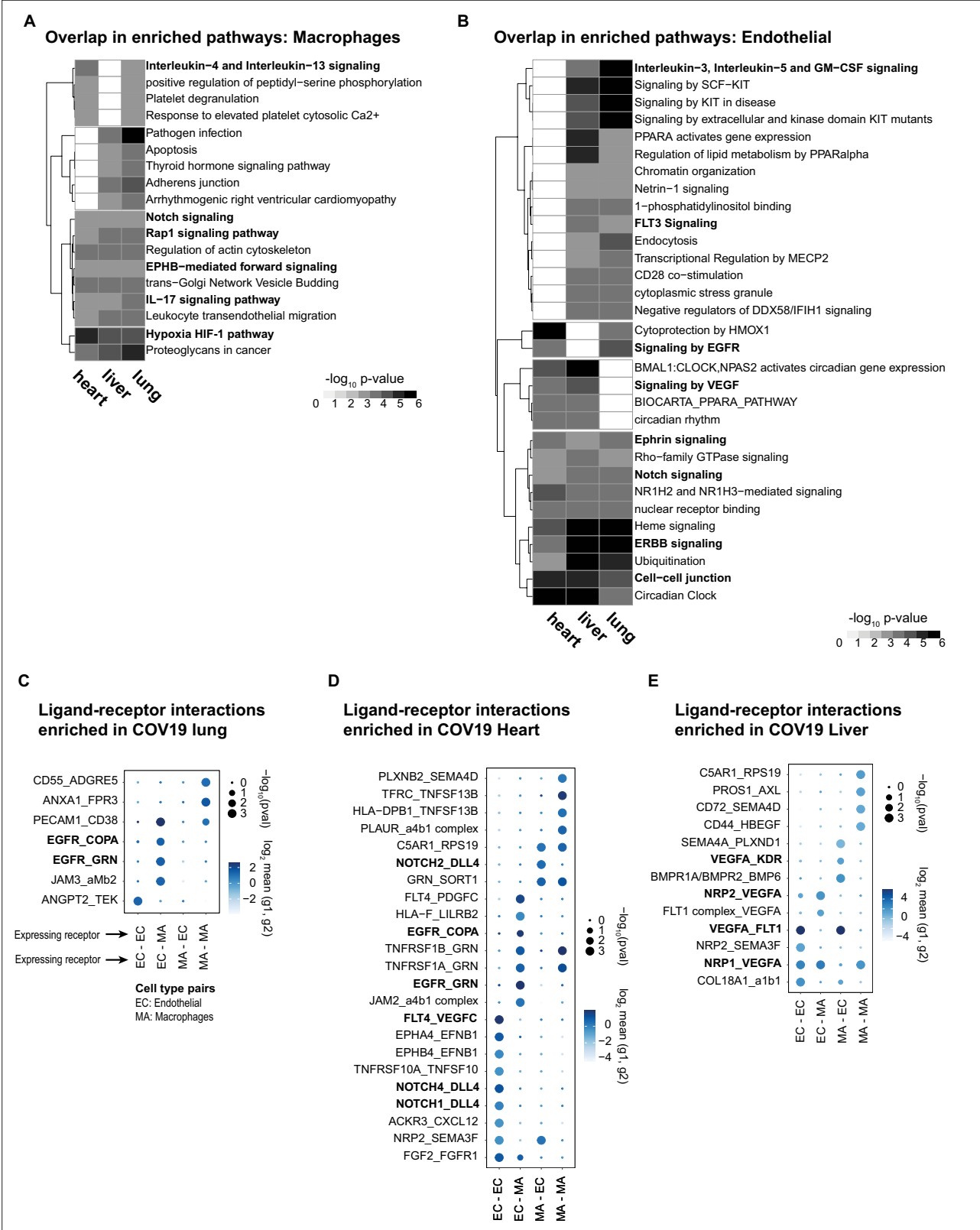

**Figure 6.** Identifying interactions between endothelial cells and macrophages. (**A**) From the shared differentially expressed (DE) genes across organs, we identified the top enriched signaling pathways for COVID-19 macrophages across the lung, liver, and heart. The value in the heatmap is the log10 p-value for the gene pathway. Only pathways with fold enrichment >3 and adjusted p-value <1e-3 in at least two organs are shown. (**B**) Enriched pathways in the shared transcriptional response of endothelial cells across the lung, liver, and heart (using the same significance thresholds as (A)). (**C–E**) Enriched

*Figure 6 continued on next page*

*Figure 6 continued*

expression of ligand-receptor components in COVID-19 macrophages and endothelial cells in the lung (**C**), heart (**D**), and liver (**E**). The x-axis indicates the pair of cell types considered (EC endothelial cells, MA macrophages). The y-axis indicates all the enriched signaling interactions found, and the circles indicate the significance and magnitude of enrichment. We calculated enrichment using CellPhoneDB on the raw sequencing counts. Only ligand-receptor pairs with adjusted p-value <1e-3 are shown.

organ-specific but rather part of a systemic response of endothelial cells and, therefore, of relevance for the development of treatments and as potential drug targets.

## Macrophage-endothelial signaling interactions in COVID-19 tissues

The enrichment of key cell-to-cell pathways such as Notch and Ephrin in the STR of endothelial cells and macrophages due to COVID-19 suggests that these two cell types may be signaling to each other. Therefore, we used CellPhoneDB (*Efremova et al., 2020*) to investigate potential signaling interactions between these two cell types by finding over-represented expression ligand-receptor pairs in COVID-19 samples compared to healthy donors (Methods).

Multiple enriched ligand-receptor pairs were identified between macrophages and endothelial cells in all three organs from COVID-19 autopsies (23 ligand-receptor pairs in the heart, 13 in the liver, and 7 in the lung, p<1e-2; *Figure 6C–E*). Among the top signaling interactions, we found expression of VEGF ligand-receptor pairs in the liver and heart (FLT4:VEGFC; VEGFA: KDR; NRP2:VEGFA; VEGFA:FLT1). In the lung, we found expression of EGFR in endothelial cells and expression of COPA and GRN in macrophages, suggesting another mechanism of cell-cell signaling between these two cell types. In the heart, we found multiple Notch ligand-receptor enriched pairs involving the expression of the Dll4 ligand in endothelial cells (*Figure 6D*). Interestingly, the expression of Notch receptors was cell-type dependent: endothelial cells expressed Notch4 and Notch1, whereas macrophages expressed Notch2 (*Figure 6D*). A Dll4-dependent signaling mechanism involving endothelial cells and macrophages in the COVID-19 heart is potentially related to HIF-1 signaling since these pathways are known to cross-talk through multiple mechanisms (*Breikaa and Lilly, 2021*; *Zheng et al., 2008*).

## Discussion

We generated the CTA, a single-cell atlas of six organs from autopsies of COVID-19 patients. Our study found that COVID-19 infection damages multiple organs, and our analyses reveal transcriptomic changes in various cell types across these organs. While the lungs are the most affected organ in COVID-19 infection, our data shows widespread transcriptional changes related to cell signaling across multiple organs and cell types (*Figure 5*). Notably, we localized these signaling changes in two affected organs: the liver and heart, where we identified dysregulated insulin and HIF signaling and prominent potential macrophage-endothelial interactions (*Figure 6*).

Collecting multiple organs from the same donors enabled cross-organ analysis of transcriptional responses. While some single-cell atlases have collected multiple organs in the past (*Jones et al., 2022*), there haven't been many efforts to do it in the context of disease (*Delorey et al., 2021*). Here, we present a cross-organ analysis of DE in the context of COVID-19. We identified a coordinated signature in macrophages and endothelial cells by analyzing the transcriptional changes in multiple organs, which is often overlooked in other studies. These coordinated signatures between macrophages and endothelial cells may be mediated by the known effects of the dysregulated immune system in the context and sequelae of COVID-19 infection.

One limitation of our study is the limited number of high-quality sequenced cells. Despite initially targeting 1 million nuclei, most samples did not pass quality control. Another consideration is that our atlas is composed of single-nuclei samples, impeding the observation of some transcriptional effects in the cytoplasm due to technological limitations. However, our dataset should be complementary to the current body of knowledge regarding COVID-19, and we expect that future analyses, including multiple datasets (*Zhang and Li, 2022*), will overcome these limitations.

The effects of COVID-19 on the human body are yet to be fully understood, and we need comprehensive maps of the changes at the transcriptional and proteomic levels. The CTA and the corresponding analyses represent an integrated effort toward understanding the effects of this disease from an organism-wide point of view (*Zhang and Li, 2022*). More generally, we expect some of

the computational analysis presented in this study to be generalized to other cell atlas datasets to reveal systemic transcriptional signatures of disease by analyzing the responses of individual cells while considering the global context of the human body. Our findings have important implications for understanding the long-term effects of COVID-19 on various organs and the increased risk of diseases linked to COVID-19 infection. For example, insulin signaling dysregulation may contribute to developing diabetes in COVID-19 patients (*Mishra and Dey, 2021*). Long-COVID, which appears to be a complex set of symptoms with variable organ dysfunction (*Morrow et al., 2022*), may also be informed by our understanding of cellular changes across multiple tissues.

The CTA represents a valuable resource that will enable further analysis, particularly cross-organ comparisons. Our data is fully available to the community through an interactive web portal from which users can directly explore the transcriptional signatures or download the count matrices for in-depth analysis. Overall, the CTA contributes to our molecular understanding of the effects of severe SARS-CoV2 infection across multiple organs and cell types.

## Additional information

### Group author details

**The COVID Tissue Atlas Consortium**
**Alejandro Granados**: Chan Zuckerberg Biohub, San Francisco, United States; **Franklin W Huang**: Department of Medicine and Liver Center, University of California San Francisco, San Francisco, United States; Department of Medicine, San Francisco Veterans Affairs Medical Center, University of California San Francisco, San Francisco, United States; **Guo N Huang**: Department of Physiology and Cardiovascular Research Institute, University of California San Francisco, San Francisco, United States; **Michael G Kattah**: Department of Medicine and Liver Center, University of California San Francisco, San Francisco, United States; **Tien Peng**: Department of Medicine and Liver Center, University of California San Francisco, San Francisco, United States; **Andreas Keller**: Department of Clinical Bioinformatics, Saarland University, Saarbrucken, Germany; **Angela Oliveira Pisco**: Chan Zuckerberg Biohub, San Francisco, United States; **Norma Neff**: Chan Zuckerberg Biohub, San Francisco, United States; **Bruce Wang**: Department of Medicine and Liver Center, University of California San Francisco, San Francisco, United States; **Hanbing Song**: Department of Medicine, San Francisco Veterans Affairs Medical Center, University of California San Francisco, San Francisco, United States; **Simon Bucher**: Department of Medicine and Liver Center, University of California San Francisco, San Francisco, United States; **Ann T Chen**: Department of Physiology and Cardiovascular Research Institute, University of California San Francisco, San Francisco, United States; **Aditi Agrawal**: Chan Zuckerberg Biohub, San Francisco, United States; **Nancy Allen**: Department of Medicine and Liver Center, University of California San Francisco, San Francisco, United States; **Benjamin Hyams**: Department of Medicine and Liver Center, University of California San Francisco, San Francisco, United States; **Deviana Burhan**; **Angela Detweiler**: Chan Zuckerberg Biohub, San Francisco, United States; **Shelly Huynh**: Chan Zuckerberg Biohub, San Francisco, United States; **Nicole Ludwig**: Department of Human Genetics, Saarland University, Homburg, Germany; **Maurizio Morri**: Chan Zuckerberg Biohub, San Francisco, United States; **Walter Schultz-Schaeffer**: Department of Neuropathology, Saarland University Medical Center, Homburg, Germany; **Michelle Tan**: Chan Zuckerberg Biohub, San Francisco, United States; **Hannah NW Weinstein**; **Rose Yan**: Chan Zuckerberg Biohub, San Francisco, United States; **Honey Mekonen**: Chan Zuckerberg Biohub, San Francisco, United States; **Rose Jia Yan**: Chan Zuckerberg Biohub, San Francisco, United States; **Aaron McGeever**: Chan Zuckerberg Biohub, San Francisco, United States; **Xiaoxin Chen**: Department of Physiology and Cardiovascular Research Institute, University of California San Francisco, San Francisco, United States; **Francisco Galdos**; **Elvira Mennillo**; **Abhishek Murti**; **Poorvi Rao**; **Lulia Rusu**; **Jamie Xie**; **Jonathan Liu**: Chan Zuckerberg Biohub, San Francisco, United States; **Sharon S Huang**: Chan Zuckerberg Biohub, San Francisco, United States; **Alexander Tarashansky**: Chan Zuckerberg Biohub, San Francisco, United States; **Kyle Awayan**: Chan Zuckerberg Biohub, San Francisco, United States

## Funding

| Funder | Grant reference number | Author |
|---|---|---|
| Chan Zuckerberg Initiative | Covid Tissue Atlas | Alejandro A Granados |
| National Institutes of Health | R01HL155622 | Tien Peng |
| National Institutes of Health | R01HL142552 | Tien Peng |

The funders had no role in study design, data collection and interpretation, or the decision to submit the work for publication.

## Author contributions

The COVID Tissue Atlas Consortium, Conceptualization, Funding acquisition, Project administration, Supervision, Writing – original draft, Writing – review and editing; Alejandro A Granados, Conceptualization, Data curation, Investigation, Software, Supervision, Visualization, Writing – original draft; Simon Bucher, Conceptualization, Data curation, Methodology; Hanbing Song, Conceptualization, Software, Validation, Investigation, Visualization, Writing – review and editing; Aditi Agrawal, Conceptualization, Supervision, Methodology; Ann T Chen, Conceptualization, Data curation, Software, Investigation, Visualization; Tien Peng, Supervision; Norma Neff, Franklin Huang, Conceptualization, Data curation, Funding acquisition, Investigation; Angela Oliveira Pisco, Conceptualization, Data curation, Funding acquisition, Investigation, Project administration; Bruce Wang, Conceptualization, Data curation, Funding acquisition, Investigation, Methodology, Project administration, Writing – original draft

## Author ORCIDs

Alejandro A Granados ![ORCID] https://orcid.org/0000-0002-6275-9800
Angela Oliveira Pisco ![ORCID] https://orcid.org/0000-0003-0142-2355
Franklin Huang ![ORCID] https://orcid.org/0000-0001-5447-0436
Bruce Wang ![ORCID] https://orcid.org/0000-0003-3486-5494

## Ethics

Organs from post-mortem control individuals and patients with COVID-19 were obtained at the time of autopsy from the University of California, San Francisco Medical Center, and the Saarland University Hospital Institute for Neuropathology. In both institutions, autopsies are exempt from local Institutional Review Board and Committee on Human Research oversight. All autopsies performed have informed consent from legal next of kin for use of tissues for research purposes.

## Decision letter and Author response

Decision letter https://doi.org/10.7554/eLife.81090.sa1
Author response https://doi.org/10.7554/eLife.81090.sa2

# Additional files

## Supplementary files

- Supplementary file 1. Patient characteristics.

- Supplementary file 2. Differential gene expression COVID-19 vs. healthy across all cell types in the COVID Tissue Atlas (CTA).

- Supplementary file 3. Pathway enrichment analysis for all cell types in COVID Tissue Atlas (CTA).

- Supplementary file 4. Shared DGE (differential gene expression) signatures in macrophages and endothelial.

- Supplementary file 5. Upregulated pathways in the shared transcriptional response of macrophages.

- Supplementary file 6. Upregulated pathways in the shared transcriptional response of endothelial cells.

- Supplementary file 7. Shared transcriptional signatures in macrophages and endothelial cells in the Broad atlas and their overlap with the COVID Tissue Atlas (CTA).

- Transparent reporting form

## Data availability

All data generated or analyzed in this study is available through the COVID Tissue Atlas (CTA) portal website. All code used in this study, including Jupyter notebooks for pre-processing, analysis, and visualization, is available on the CTA GitHub repository (copy archived at *Granados and Pisco, 2023*).

The following dataset was generated:

| Author(s) | Year | Dataset title | Dataset URL | Database and Identifier |
|---|---|---|---|---|
| COVID Tissue Atlas Consortium | 2022 | COVID Tissue Atlas | https://covid-tissue-atlas.ds.czbiohub.org | CTA, covid-tissue |

The following previously published dataset was used:

| Author(s) | Year | Dataset title | Dataset URL | Database and Identifier |
|---|---|---|---|---|
| Regev A | 2021 | COVID-19 tissue atlases reveal SARS-COV-2 pathology and cellular targets | https://www.ncbi.nlm.nih.gov/geo/query/acc.cgi?acc=GSE171668 | NCBI Gene Expression Omnibus, GSE171668 |

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

## Appendix 1

### Details for the COVID Tissue Atlas Consortium

Overall Project Direction and Coordination

Alejandro Granados, Franklin W Huang, Guo N Huang, Michael G Kattah, Tien Peng, Andreas Keller, Angela Oliveira Pisco, Norma Neff, Bruce Wang

Writing group

Alejandro Granados, Hanbing Song, Simon Bucher, Ann T Chen, Angela Oliveira Pisco, Norma Neff, Franklin W. Huang, Bruce Wang

Organ Processing and Library Preparation

Aditi Agrawal, Nancy Allen, Benjamin Hyams Simon Bucher, Deviana Burhan, Angela Detweiler, Shelly Huynh, Nicole Ludwig, Maurizio Morri, Walter Schultz-Schaeffer, Michelle Tan, Hannah NW Weinstein, Rose Yan

Sequencing

Norma Neff, Michelle Tan, Angela Detweiler, Honey Mekonen, Rose (Jia) Yan

Data processing

Aaron McGeever, Angela Oliveira Pisco, Alejandro Granados

Cell Type Annotation Expert Group

Nancy Allen, Xiaoxin Chen, Francisco Galdos, Alejandro Granados, Guo N Huang, Michael G Kattah, Elvira Mennillo, Abhishek Murti, Poorvi Rao, Lulia Rusu, Hanbing Song, Tien Peng, Bruce Wang, Jamie Xie

Data Analysis

Alejandro Granados, Ann T Chen, Hanbing Song, Jonathan Liu, Abhishek Murti

Data release and portal

Alejandro Granados, Sharon S Huang, Alexander Tarashansky, Angela Oliveira Pisco, Kyle Awayan

