## [Editor Report]

This important work provides a valuable data resource to study the systemic effects of severe COVID-19 infection. It further provides compelling evidence for a conserved transcriptional signature in macrophages and endothelial cells in response to COVID-19 and suggests potential molecular interactions between the two cell types. The work will be of broad interest to researchers investigating the physiological impacts of viral infection and their potential treatment.

---

## [Decision Letter]

**Decision letter after peer review:**

Thank you for submitting your article "Single nuclei characterization of pervasive transcriptional signatures across organs in response to COVID19" for consideration by *eLife*. Your article has been reviewed by 2 peer reviewers, and the evaluation has been overseen by a Reviewing Editor and Aleksandra Walczak as the Senior Editor. The reviewers have opted to remain anonymous.

Essential revisions:

Please provide more evidence to support the presence of ABI cells in the data.

Please clarify the presence and potential role of neuronal cells in the datasets studied, and their fate during infection.

Please indicate whether the increase of endothelial cells is related to blood vessel formation, e.g. through a cell subtype analysis. A higher resolution study of these subtypes and their potential role will add value to this resource.

Please describe other variables that might be associated with changes in cell proportions and gene expression, like age and comorbidities.

Please clarify the potential biases and limitations of this study, and how it compares with other previous similar studies, emphasizing the novel insights and/or complementarity of this study.

Please clarify the nature of the coordinated response put forward in the manuscript. In particular, whether this is consistent across cell types, or specific to some of them, or not. Any orthogonal validation that could be used to confirm these claims will add further support to the principles put forward in the manuscript

*Reviewer #1 (Recommendations for the authors):*

The authors performed scRNA-seq of six human organs across 20 autopsies. The data quality and batch effects were well controlled in the analysis. Based on this, the authors identified specific and common changes in each organ after infection with SARS-CoV2. Specifically, the authors observed dysregulation in insulin and HIF signal pathways in the heart and lung. In addition to lung, the authors found significant transcriptional changes in macrophages and endothelial cells in heart and liver. This study provides a valuable data resource to study the systemic effects of severe COVID-19. However, it needs to address the following issues:

1. The anchor integration alone was not sufficient to indicate the presence of ABIs cells in the data, did the authors observe the expression of ABIs cells markers in the cells they predicted as ABIs in their data?

2. Considering that the nervous system may be affected by COVID-19 autopsies, did the authors detect neuronal cells in different organs? What happens to them after infection with SARS-CoV2?

3. The analysis showed a much higher proportion of endothelial cells in the lungs and heart in COVID-19 than in control, were increased blood vessels observed in these organs? Do these extra vessels function normally? If yes, it would be interesting to explore why these organs need so many endothelial cells.

4. Whether alterations in cell proportions and gene expression in COVID-19 are associated with age or some underlying diseases？

*Reviewer #2 (Recommendations for the authors):*

Expansion of hepatocytes in the liver is very interesting. Hepatocytes play a significant role in both glucose and lipid metabolism. These 2 pathways are tightly regulated by insulin signalling. Since the authors are reporting dysregulation of insulin signalling with potential hepatocytes regeneration caused by COVID-19, this could be further discussed. For example, did changes of insulin signalling lead to dysregulation of lipid metabolism?

Differential abundance analysis (e.g. Milo) is a better alternative than plotting % of cell types in health vs COVID-19.

The term COVID-19 has been inconsistently labelled throughout the paper, COVID-19, COVID, Cov19. It'd be great to have the term used consistently throughout the paper to avoid confusion.

---

## [Author Response]

Essential revisions:Please provide more evidence to support the presence of ABI cells in the data.

We thank the reviewer for the advice and agree that it is essential to provide more evidence of ABI cells’ existence in our dataset. Therefore, we computed the top differentially expressed genes of the ABI (transitional AT2) cells from the Habermann et al. dataset and derived an ABI signature gene set. Next, we investigated this signature in the CTA and integrated datasets (CTA + Habermann et al.). Even though no subcluster showed enrichment for the ABI signature, a number of the ABI markers, in particular, those genes expressed in the KRT17+/KrRT5- cells, appeared in our basal population (Figure 2F), suggesting that the ABI population (Kathiriya et al.) was present in the COVID-19 lung. We speculate that trans-differentiation from endogenous AT2s could account for the gain of abnormal basal cells in the alveoli. We revised the manuscript accordingly to reflect this finding. We included a new heatmap to show the expression of the ABI gene signature within the lung epithelial cells from the CTA (Figure 2F).

Similarly, we derived a signature gene set of the pre-alveolar Type 1 transitional cell state (PATS) population in COVID-19 lungs from the Broad atlas. However, even though these PATS cells bear similarities to the previously described *ABIs/Transitional AT2s/aberrant basaloid* cells from IPF lungs (Habermann et al. and Kathiriya et al.), we did not detect any specific cell cluster in the CTA dataset enriched for the PATS signature. This result suggests that the PATS population is likely patient-specific ( or specific to sequencing technology differences) and, therefore, was not detected in the CTA donors. Accordingly, in the revised manuscript, we modified our PATS analysis description.

Please clarify the presence and potential role of neuronal cells in the datasets studied, and their fate during infection.

We identified a small subset of cells in the hearts of COVID-19 donors expressing high neuronal markers levels. Among the most enriched gene markers in this cluster compared to other cell types in the heart, we found the neuron-specific genes NRX1, NRX3, and XKR4, along with cancer-related genes such as ANK3 (typically enriched in oligodendrocytes) and BCL2 (also found in skeletal myocytes). We also found CDH19 among this cluster's top markers, typically in the brain and heart muscle. Therefore, the cluster comprises a strong signature of neuron-like cells expressing heart-related genes. While interesting, it is challenging to confidently assign an identity to this cluster due to a concern regarding RNA contamination.

During our decontamination pipeline, we noticed that some of the cells expressing neuronal markers also expressed markers exclusively expressed in other cell types. We were able to fix some of this contamination. Still, we suspected that the expression of neuronal markers correlated with RNA contamination. Therefore, we decided to err on caution and not include the cluster in our downstream analysis. Finally, the Broad atlas also identified a small neuronal population in the heart of COVID-19 autopsies (Delorey, 2021), suggesting that the cluster might represent a real population. However, further investigation and deeper sequencing might be necessary to assess the existence and relevance of such a neuron-like population.

Please indicate whether the increase of endothelial cells is related to blood vessel formation, e.g. through a cell subtype analysis. A higher resolution study of these subtypes and their potential role will add value to this resource.

To address this comment, we separately applied a sub-clustering of the endothelial cell populations from the lung and the heart (Figure 3 – Supplement figures 2 and 3). After unbiased clustering, we used a list of gene markers from recently published human lung and heart endothelial cell single-cell atlases to compare with the expression profiles of different endothelial cell clusters in our data. We identified four sub-clusters of endothelial cells in the lung and 4 in the heart. We found that the overall increase in endothelial cells in both organs is driven primarily by an increased proportion of capillary endothelial cells in COVID-19 samples compared to the controls (Figure 3 – Supplement figures 2 and 3). Consistently, a gene ontology analysis from capillary endothelial cells in the lungs showed angiogenesis significantly upregulated. Therefore, our data indicate that capillary endothelial cells are in COVID-19 infection. We attempted a similar analysis on our liver endothelial cell data, but due to the much smaller cell numbers encountered significant patient-specific batch effects that precluded further analysis (data not shown).

We included two supplemental figures to investigate the changes in endothelial populations. Our results are preliminary but suggest a pivotal role of capillary endothelial cells.

Please describe other variables that might be associated with changes in cell proportions and gene expression, like age and comorbidities.

We appreciate the comment and are interested in exploring the potential correlations between transcriptional changes and clinical variables (e.g., age, race, comorbidities, etc.). We attempted regression analysis using factors including age, race, disease severity (ICU status), and infection length but couldn’t find a strong signal for correlations with gene expression changes. Unfortunately, we don’t have enough donors for the statistical analysis to be reliable. Notably, previous studies also failed to identify correlations with clinical metadata (Delorey, 2021) due to similar issues of low sample size. While we believe those correlations exist and are crucial to understanding the disease, the current limitation is the number of donors in the dataset. By compiling data across multiple datasets, we anticipate that future studies will be able to explore these questions in more detail.

Please clarify the potential biases and limitations of this study, and how it compares with other previous similar studies, emphasizing the novel insights and/or complementarity of this study.

The main novelty of this study is the collection of multiple organs from the same donors and the cross-organ analysis of transcriptional responses. This feature allowed us to identify a coordinated transcriptional response in macrophages and endothelial cells by analyzing the transcriptional changes in multiple organs, which is often overlooked in other studies. We further explored these changes by identifying potential molecular interactions between these two cell types.

In line with the CZ Biohub commitment to open science, our data is available for exploration through a web portal and can be easily accessed by the community.

The main limitations of our study include the number of high-quality cells in the analysis, which is lower than in other studies. We sequenced up to 1 million nuclei for this project, but less than 10% passed our quality control. In addition, we believe a significant cause is the longer time between patient death and organ harvest for our samples, with an average time of 63 hours after death, with a few specimens collected more than 96 hours after death. Since our atlas is composed of single-nuclei data, another limitation is the potential transcriptional effects that we cannot observe due to technological limitations. However, our dataset should be complementary to the current body of knowledge regarding COVID-19, and we expect that future analyses, including multiple datasets, will overcome this limitation.

A general bias we are aware of is the lack of race and sex diversity in our donors: 17 males vs. three females, and ethnicities Hispanic (n=5), African American (n=2), Asian (n=1), and White (n=12). Therefore, our results and conclusions are limited to the context of our donor population.

We extended the discussion in the main text to include some of the novelties and limitations of the study.

Please clarify the nature of the coordinated response put forward in the manuscript. In particular, whether this is consistent across cell types, or specific to some of them, or not. Any orthogonal validation that could be used to confirm these claims will add further support to the principles put forward in the manuscript

Orthogonal validation of the coordinated response would strengthen the validity of our results. *The coordinated response is strong with only two cell types -> say that we tried.*

We, therefore, applied our analysis to the COVID-19 dataset from the Broad Institute (Delorey et al., 2021). The Broad’s single-nuclei atlas comprises samples from hearts, livers, and lungs from COVID-19-infected autopsies.

Our main finding in the CTA is the coordinated transcriptional response in macrophages and endothelial cells across multiple organs, particularly in the heart, liver, and lung. Therefore, the Broad atlas provides an ideal opportunity for independent validation of our results. Here is a summary of the main results:

By applying the same analysis to the Broad atlas, we identified a coordinated response in macrophages and endothelial cells across the heart, liver, and lung of COVID-19 autopsies.The list of genes showing a coordinated response in the Broad atlas showed high overlap with the coordinated response found in the CTA (50% overlap in macrophages and 55% overlap in endothelial cells).Further, we found a strong correlation between the log-fold changes in gene expression for the overlapping set of genes. And we included scatterplots showing these correlations in Figure 5.We explain in detail the analysis in the text and emphasize the importance of this external validation.

The additional panels and text strongly support our main finding of a coordinated transcriptional response in macrophages and endothelial cells across multiple organs in response to COVID-19.

Reviewer #1 (Recommendations for the authors):The authors performed scRNA-seq of six human organs across 20 autopsies. The data quality and batch effects were well controlled in the analysis. Based on this, the authors identified specific and common changes in each organ after infection with SARS-CoV2. Specifically, the authors observed dysregulation in insulin and HIF signal pathways in the heart and lung. In addition to lung, the authors found significant transcriptional changes in macrophages and endothelial cells in heart and liver. This study provides a valuable data resource to study the systemic effects of severe COVID-19. However, it needs to address the following issues:1. The anchor integration alone was not sufficient to indicate the presence of ABIs cells in the data, did the authors observe the expression of ABIs cells markers in the cells they predicted as ABIs in their data?

See the reply to Essential revisions comment #1 above.

2. Considering that the nervous system may be affected by COVID-19 autopsies, did the authors detect neuronal cells in different organs? What happens to them after infection with SARS-CoV2?

See the reply to Essential revisions comment #2 above.

3. The analysis showed a much higher proportion of endothelial cells in the lungs and heart in COVID-19 than in control, were increased blood vessels observed in these organs? Do these extra vessels function normally? If yes, it would be interesting to explore why these organs need so many endothelial cells.

See the reply to Essential revisions comment #3 above.

4. Whether alterations in cell proportions and gene expression in COVID-19 are associated with age or some underlying diseases？

See the reply to Essential revisions comment #4 above.

Reviewer #2 (Recommendations for the authors):Expansion of hepatocytes in the liver is very interesting. Hepatocytes play a significant role in both glucose and lipid metabolism. These 2 pathways are tightly regulated by insulin signalling. Since the authors are reporting dysregulation of insulin signalling with potential hepatocytes regeneration caused by COVID-19, this could be further discussed. For example, did changes of insulin signalling lead to dysregulation of lipid metabolism?Differential abundance analysis (e.g. Milo) is a better alternative than plotting % of cell types in health vs COVID-19.

Thanks, we appreciate the comment, although our limited number of samples prevents us from understanding the changes quantitatively. Integration with other studies in the future will facilitate the investigation of changes in cell populations.

The term COVID-19 has been inconsistently labelled throughout the paper, COVID-19, COVID, Cov19. It'd be great to have the term used consistently throughout the paper to avoid confusion.

Thanks. We reviewed the manuscript and edited it where needed.